# Single Cell Protein Production through Multi Food-Waste Substrate Fermentation

**Alessia Tropea** [1,*], **Antonio Ferracane** [2,*] , **Ambrogina Albergamo** [3], **Angela Giorgia Potortì** [3] ,
**Vincenzo Lo Turco** [3] and **Giuseppa Di Bella** [3]

1   Department of Research and Internationalization, University of Messina, Via Consolato del Mare, 41, 98100 Messina, Italy
2   Department of Chemical, Biological, Pharmaceutical and Environmental Sciences, University of Messina, Polo Annunziata, Viale Annunziata, 98166 Messina, Italy
3   BioMorf Department, University of Messina, Viale Annunziata, Polo Universitario, 98168 Messina, Italy; aalbergamo@unime.it (A.A.); agpotorti@unime.it (A.G.P.); vloturco@unime.it (V.L.T.); gdibella@unime.it (G.D.B.)
*   Correspondence: atropea@unime.it (A.T.); aferracane@unime.it (A.F.)

**Abstract:** Today, food valorization represents an important challenge to environmental sustainability. Food waste can be used as a substrate for single cell protein production suitable for animal feed. In this study, animal and agricultural food waste, represented by fish, pineapple, banana, apple, and citrus peels, have been used simultaneously as a fermentation substrate for single cell protein production by *Saccharomyces cerevisiae*, to evaluate the possibility of using a multi complex substrate for a simultaneous biovalorization of different food waste. The fermentation process was implemented by the supplementation of a hydrolytic enzyme and nutrient to allow the best yeast growing conditions. At the end of the process, the final substrate was enriched in protein, reaching up to 40.19% of protein, making the multisubstrate useful for animal feed. The substrate was also investigated for crude lipid, ash, lignin, soluble and insoluble sugar. The substrate composition at the end of the fermentation process was represented by 14.46% of crude lipid, 1.08% ash, 6.29% lignin. Conversely, the soluble and insoluble sugars dropped down from 20.5% to 6.10% and 19.15% to 2.14%, respectively, at the end of the process.

**Keywords:** fermentation; single cell protein; food waste; *Saccharomyces cerevisiae*; sustainability; agrifood waste; fish waste

## 1. Introduction

Food waste is becoming an increasingly important issue at both local and global levels [1]. The global intensification of food production is generating significant agricultural waste [2]. The Food and Agriculture Organisation of the United Nations (FAO) has estimated that one third of all food production is lost or wasted globally, equivalent to 1.3 billion tons of food produced for human consumption wasted each year with an economic loss of EUR 800 billion [3]. About 44–47% is represented by vegetable, fruit, fish and meat produced every year and wasted [4]. Normally, most of these wastes are incinerated or go to landfill sites [5]. The improper management of these wastes can constitute a public health risk and cause environmental problems, such as diseases and air pollution [5,6]. Appropriate waste management is recognized as an essential prerequisite for sustainable development, contributing to the attainment of the global sustainability goals (SDGs 12 and 13).

From a chemical point of view, food waste is mainly composed by carbohydrate polymers such as starch, proteins, lipids, cellulose, and other microelements [7–10]. Due to this composition, it is classified as a low cost, high potency second generation feedstock [1], and usable as a substrate for microbial fermentation for bioconversion into value added

products, such as enzymes, feed additives, biofuels, animal feeds as well as other useful chemicals or products, food grade pigments, and single cell protein (SCP), enhancing food security and environmentally sustainable development [11–21].

The production of SCP via fermentation is a biochemical process carried out by microorganisms such as yeast, bacteria, and fungi, which breakdown complex substrates into simpler compounds for growing [22]. Feedstock and waste used in SCP production are mostly represented by agricultural sources and the product can be used as a protein supplement in either food or feed [23,24].

Single cell protein technology is designed to solve worldwide protein shortage [25,26] and has shown a great advantage because it is independent of climate, soil characteristics and, not least, on available land [27].

Previous studies refer to SCP production via fermentation by testing a single kind of food processing or agricultural waste each time, such as wheat bran [28], orange and lemon peel [29,30] banana peel [31,32], fish waste [16], pineapple waste [23,24], whey [33] and others.

According to Anupama and Ravindra [34], fermented products containing SCP can feed both humans and animals, thereby replacing expensive conventional sources of protein such as fishmeal and soymeal. However, it is important to use a microorganism that are "generally regarded as safe (GRAS)" for fermentation in order to promote its use [35–39]. The yeast *Saccharomyces cerevisiae* is widely accepted, considering its use in traditional fermentation and its nutritional quality [22]. Moreover, due to its immunostimulating compounds, such as nucleic acid, b-glucans and mannan oligosaccharides [40,41], it has been used, e.g., for aquafeed production by fish waste fermentation, after medium supplementation with lemon peel used as natural filler [16] for reducing natural liquefaction occurring during fish waste fermentation. This study has evaluated the possibility of using a multicomplex substrate, made up of fish waste and different agricultural wastes, to set up a fermentation process, carried out by *Saccharomyces cerevisiae* ATCC 36858, which allows the simultaneous biovalorization of different food wastes without the necessity to separate the waste coming out from food process to obtain a single cell protein suitable as feed. Considering the chemical composition of the waste used in this study, the medium was supplemented with cellulolytic enzymes, carrying out a simultaneous saccharification and fermentation process, in order to obtain free sugars from the agricultural food waste cell wall suitable for the yeast growth. The fermentation medium was also investigated for the proximate composition, with a particular focus on the soluble and insoluble sugars resulting from the cell wall waste hydrolysis.

## 2. Materials and Methods

### 2.1. Substrate

Fish wastes, represented by head, viscera, skin and bones, pineapple, banana, apple, and citrus peels were provided by local companies. Samples were collected directly from the companies, forwarded to the laboratory under refrigerated condition and stored at −20 °C until tests were performed. Wastes, in a proportion of 20% (*w/w*) for each variety (1:5 *w/w*), were cut into small pieces and homogenized in a blender for 5 min.

### 2.2. Microorganism

*Saccharomyces cerevisiae* ATCC 36858 (Manassas, VA, USA) was cultured and maintained on yeast medium (YM) agar (yeast extract 3 g/L, malt extract 3 g/L, peptone 5 g/L, glucose 10 g/L, agar 20 g/L; Oxoid, Basingstoke, UK) at 4 °C. To carry out the tests, *S. cerevisiae* was cultured overnight at 30 °C on a rotary shaker (INNOVA 44, Incubator Shaker Series, New Brunswick Scientific, Edison, NJ, USA) at 200 rpm, in tubes containing 20 mL YM.

After overnight incubation, cell suspensions were aseptically harvested by centrifugation (3000 rpm, 5 min, Centrifuge 5810 R, Eppendorf UK Ltd., Stevenage, UK), the

supernatant (YM media) discarded, and the yeast cells rinsed twice in 5 mL 0.9% (*w/v*) NaCl to minimize nutrient transfer from seed culture to fermentation medium.

The total viable yeast cells were measured by using a cell count reader (Nucleocounter YC 100™, ChemoMetec, Allerød, Denmark). The standard yeast culture contained $10^8$ cells per mL of *S. cerevisiae* ATCC 36858.

### 2.3. Experimental Set-Up

Fermentation tests were carried out in a 5 L batch fermenter (Biostat Biotech B, Sartorius Stedim Biotech, Goettingen, Germany). The fermenter was equipped with one four-bladed Rushton turbine and the usual control systems: temperature, pH, $pO_2$ and a foam detector.

Food wastes were homogenized in a blender. The resulting homogenate, with a dry matter content of 37% (*w/w*), was diluted with water to a 15% dry matter, in a working volume of 3.5 L and sterilized for 15 min at 121 °C.

According with Tropea et al. [38] the fermentation medium was supplemented with urea phosphate salt 2.3 g/L; KCl 0.2 g/L; $MgSO_4 \cdot 7H_2O$ 3.8 g/L; Ca-pantothenate 0.0833 mg/L; biotin 0.0833 mg/L. Moreover, the fermentation medium was supplemented with 1mL/L antifoam Sigma 289 (Merck KGaA, Darmstadt, Germany).

The process was carried out in simultaneous saccharification and fermentation mode. According to previous literature [42,43] reporting the capacity of the enzyme used in this work to carry out the hydrolysis at $30 \pm 2$ °C, the fermentation parameters were selected as the best conditions for *S. cerevisiae* growth [44,45]: 30 °C, airflow 0.5 L/min, pH 4.5 and constant stirring at 300 rpm. The pH value was previously adjusted from 3.5 up to 4.5, using 2 M NaOH.

The fermentation substrate was degraded to convert cellulose content, derived from agricultural waste, into available sugars, by enzymatic treatment, using 10 filter paper units (FPU) per gram substrate of cellulase (Cellic® CTec2 Novozymes Corp, Bagsvaerd, Denmark).

Duplicate broth samples were withdrawn from the reaction vessel using a 20 mL syringe: all the samples for the analytical determinations were heated at 100 °C for 10 min to inactivate the enzyme and stop any further fermentation. Samples for protein content and biomass determination were centrifuged and rinsed twice with 0.9% NaCl and once with demineralized water and, finally, freeze dried prior to analysis.

Samples for the other determinations were frozen at −18 °C until analyzed. Throughout fermentation, the pH was maintained at 4.5 by automatic feeding of ammonia.

### 2.4. Chemicals

Chemicals were provided by Sigma Aldrich (Bellefonte, PA, USA), excepting for galacturonic acid and glucose provided by Fluka Biochemical; KCl, $MgSO_4 \cdot 7H_2O$, and Ca-pantothenate provided by Fisher Scientific; biotin, provided by Calbiochem.

### 2.5. Crude Protein, True Protein, Moisture, Ash and Lignin Determinations

Representative samples were drained off for crude protein content testing, using the method reported by the AOAC [46].

Crude protein was determined as total N, multiplying the results for the conversion factor of 6.25, by using Büchi Kjeldahl instrument, equipped with Büchi Distillation Unit B-324, Digestion Unit K-424 and Scrubber B-414 (Büchi, Switzerland). True protein content was evaluated by Folin–Ciocalteau method, as reported by Lowry et al. [47].

The dry weights were calculated as steady weights after 2 h at 110 °C, using a Mettler PM 200 equipped with a Mettler LP16 IR balance (Mettler-Toledo GmbH, Laboratory & Weighing Technologies, Greifensee, Switzerland).

Ash determination was carried out according to the AOAC method [46]. Klason lignin was quantified gravimetrically [48]. Briefly, AIRs samples were dispersed in 1.5 mL of $H_2SO_4$ (w = 72%) and incubated at 30 °C for 1 h shaking. The samples were further incubated for 2.5 h after diluting with 10.5 mL water in a temperature controlled oven

set at 100 °C. The residues were recovered by filtration through preweighed sintered glass funnels (10 mm diameter, Fisher Scientific UK Ltd., Loughborough, Leicestershire, UK) under vacuum. The insoluble material was rinsed using warm water in order to eliminate the acid. The glass filters were dried at 50 °C until constant weight and Klason lignin calculated gravimetrically as a percentage of the starting material. All samples were analyzed in triplicate.

### 2.6. Alcohol Insoluble Residues (AIR)

AIRs were prepared prior to analysis for cell wall sugars. Wet fermented samples were homogenized for 1 min at max speed in a Janke & Kunnel, Ika-Werk Ultra-Turrax homogenizer at room temperature. Samples were then poured into boiling ethanol, to obtain a final mixture with EtOH concentration of 85% (*v/v*), considering the water content of the sample. A total of 50 mL of 70% EtOH to wash and collect any sample particles from the homogenizer was used. The insoluble residue remaining after this treatment was recovered by vacuum filtration through a 5 μm nylon filter NYBOLT using a Buchner funnel. After 2 further sequential extractions in boiling 85% ethanol (*v/v*), the residue was extracted in boiling absolute ethanol then washed with cold absolute ethanol. The final filtrate was dried by Büchi Rotary Evaporator at 40 °C, recovered in water and tested for residual soluble sugars. The insoluble residue was washed with 2 volumes acetone and after removal by suction, dried to constant weight at 40 °C [49,50] and analyzed for insoluble sugars determination.

### 2.7. Sugar Analysis

Insoluble sugars were released from AIR samples by hydrolysis and analyzed by gas chromatography flame ionization detection (GC-FID) after conversion to their alditol acetates. The quantification was carried out using 2-deoxyglucose as internal standard [51]. Monosaccharides were released from polysaccharides with prehydrolysis of the samples using 0.2 mL of 72% (*w/w*) $H_2SO_4$ for 3 h at room temperature followed by 2.5 h hydrolysis in 1 M $H_2SO_4$ at 100 °C. After 1 h hydrolysis, 0.5 mL were collected for uronic acids determination. After hydrolysis, the reduction and acetylation of the monosaccharides were performed, and the alditol acetates were analyzed by Shimadzu Gas Chromatograph GC-2010 equipped with a Flame Ionization Detector (GC/FID) (Kyoto, Japan) with a capillary column DB-225 (30 m length, 0.25 mm ID and 0.15 μm df, 50%-Cyanopropylphenyl-dimethylpolysiloxane) (Agilent Technologies, Folsom, CA, USA) [52].

Analysis of supernatant fractions for residual soluble sugars determination followed the same protocol but starting from a hydrolysis in 1 M $H_2SO_4$. The oven temperature program was as follows: 200 °C to 220 °C at a rate of 40° C/min (7 min), increasing to 230 °C at a rate of 20 °C/min (1 min). The temperature of injector was 220 °C and the detector was 230 °C. Carrier gas was hydrogen, and the flow rate was set at 1.7 mL/min. The free sugars were identified and quantified based on their retention times and response factors obtained by injection of standards. Uronic acid content was determined by the m-phenylphenol colorimetric method [53] modified by Rae et al. [54], using galacturonic acid as standard. To the 0.5 mL of diluted hydrolyzed sample (1:4) was added 3 mL of boric acid 50 mM $H_2SO_4$ 98% (*w/w*), after shaken the test tubes were heated at 100 °C during 10 min. After cooling, 100 μL of m-phenylphenol was added, reacting 30 min in dark, and the absorbance was measured at 520 nm. All samples were analyzed in triplicate.

### 2.8. Crude Fat and Fatty Acid Determination

Samples were extracted with a mixture of chloroform and methanol (2:1). The mixture was allowed to stand overnight and the lower lipid layer, transferred into a pretreated and weighed flask, was dried off. The difference in the two weights gave the weight of the fat [16].

The fatty acid analysis was performed by gas chromatography after transmethylation with 2% $H_2SO_4$ in methanol at 80 °C for 10 min. The separation and quantification of

fatty acid methyl esters were elucidated by a gas chromatograph (GC) equipped with a split/splitless injector and a flame ionization detector (FID) (Dani Master GC1000, Dani Instrument, Milan, Italy). A capillary column Supelco SLB-IL100, (60 m × 0.25 mm ID, 0.20 μm film thickness) (Supelco, Sigma Aldrich, USA), was employed. The following experimental conditions were used: injector temperature 220 °C; oven temperature from 130 °C to 210 °C (10 min holding) at a rate of 2 °C/min; detector temperature 240 °C; carrier gas He at constant velocity rate of 30 cm/sec; injection volume 1 μL, with a split ratio of 1:100.

Fatty acids were identified by comparing with reference standards Supelco 37 component FAME mix in methylene chloride. All samples were analyzed in triplicate.

### 2.9. Statistical Analysis

The studies of significant differences were carried out by Kruskal–Wallis tests using SPSS 13.0 software package for Windows (SPSS Inc., Chicago, IL, USA).

## 3. Results and Discussion

### 3.1. Proximate Composition

Table 1 shows the results concerning the crude and true protein, crude lipid, ash, and lignin percentage, determined during all the fermentation process. All statistical evaluation were performed at $\alpha$ = 0.05.

**Table 1.** Crude and true protein, crude lipid, ash, and lignin percentage at different fermentation time.

|  | Crude Protein % | True Protein % | Crude Lipid % | Ash % | Lignin % |
|---|---|---|---|---|---|
| 0 h | 8.52 ± 0.81 (A) | 6.06 ± 0.18 (A) | 11.47 ± 0.82 (A) | 3.93 ± 0.11 (A) | 1.65 ± 0.03 (A) |
| 24 h | 16.56 ± 0.95 (A) | 14.66 ± 0.73 (A) | 12.57 ± 0.90 (A) | 3.90 ± 0.14 (A) | 2.72 ± 0.59 (A) |
| 48 h | 24.24 ± 0.93 (A) | 22.68 ± 1.09 (A) | 12.84 ± 0.99 (A) | 2.98 ± 0.16 (AB) | 4.46 ± 0.42 (AB) |
| 72 h | 35.77 ± 1.57(B) | 33.48 ± 1.32 (B) | 12.53 ± 0.76 (A) | 2.84 ± 0.22 (AB) | 4.99 ± 0.09 (AB) |
| 96 h | 39.60 ± 1.34 (B) | 37.69 ± 1.36 (B) | 14.59 ± 0.83 (A) | 1.36 ± 0.26 (B) | 6.28 ± 0.55 (B) |
| 120 h | 40.19 ± 2.13 (B) | 38.43 ± 1.37 (B) | 14.46 ± 0.83 (A) | 1.08 ± 0.08 (B) | 6.29 ± 0.61 (B) |
| 144 h | 38.38 ± 1.07 (B) | 36.55 ± 1.34 (B) | 14.36 ± 0.76 (A) | 1.09 ± 0.07 (B) | 6.32 ± 0.62 (B) |

The values in a column with different letters are significantly different ($p \leq 0.05$).

Initial crude and true protein detected in the starting material were 8.52 ± 0.81% and 6.06 ± 0.18%, respectively. Their concentration increased after 72 h, reaching up the 35.77 ± 1.57% for crude protein and 33.48 ± 1.32% for true protein. The highest protein percentage in the substrate was reached at 120 h. In fact, the crude protein percentage at that time was 40.19 ± 2.13%, corresponding to a true protein content of 38.43 ± 1.37%. As can be observed, at the end of the fermentation process, the protein percentage decreased to 38.38 ± 1.07% and 36.55 ± 1.34%, for crude and true protein, respectively. This decrease can be ascribed to the autolysis phenomena due to the yeast cells deaths during the decline phase of the growth curve [55,56]. The protein percentage increased during the substrate fermentation by the yeast, allowing the obtainment of a substrate enriched in protein, having a suitable percentage for aquafeed [16,22] and for feed [30,57]. Moreover, comparing these results with previous studies where the fermentation medium was represented by a single variety of waste per time, it is possible to notice that a higher percentage of protein was reached using a multi complex medium. SCP production from pineapple waste, investigated by Aruna and Tropea et al. [21,58], allowed the obtainment of the highest crude protein yield, of 13.56% and 17.2%, respectively, whereas the true protein percentage was 10.86%. In another study, where selected food wastes (banana peel, citrus peel, carrot pomace and potato peel) were investigated separately for SCP production, it was pointed out that maximum SCP yield was obtained at 30 °C for 7 days reaching a percentage lower than 8% for citrus and banana peels [57]. Whereas a higher protein percentage, 48.55%, was reached in a previous study, where fish waste was used simultaneously with citrus peel [16] in a waste proportion of 2:1 (*w/w*).

The crude lipid percentage determined on the initial substrate was 11.47 ± 0.82%. During all the process, this value did not increase significantly, reaching up at the end of the fermentation just 14.36 ± 0.76%, according with Tropea et al. [16]. As reported in literature, the crude lipid increasing is mainly due to the yeast growth when the substrate is lacking in nitrogen supplementation [59]. In this study, the urea phosphate supplementation allowed to minimize the lipid production for maximizing the protein increasing during the fermentation process, in respect of the considerable effects on *S. cerevisiae* growth rate exhibit by the culture medium [30,60].

Ash and lignin, whose percentages are reported in Table 1, in the starting material were 3.93 ± 0.11% and 1.65 ± 0.03% respectively. Conversely, the amounts detectable at the end of the fermentation process in the fermented material were 1.09 ± 0.07% and 6.32 ± 0.62%, respectively. The lignin amount increasing can be explained by remembering that lignin cannot be fermented by the yeast [61]; so, according to the literature, its increasing on dry matter bases is typically due to enzymatic fiber hydrolysis [62]. During all the fermentation process, the ash percentage was decreasing from 3.93 ± 0.11% down to 1.09 ± 0.07%. This trend, as already stated by Tropea et al. [16], can be ascribed to partial ash utilization by the yeast as source of minerals [63].

### 3.2. Fatty Acid Composition

Table 2 shows the results regarding the fatty acid contents at different fermentation times. As can be observed, at the beginning of the fermentation the saturated fatty acids (SFAs) were detected in a lower percentage, 24.91 ± 1.51%, in comparison with monounsaturated fatty acids (MUFAs) and polyunsaturated fatty acids (PUFAs), with these showing a percentage of 35.48 ± 2.04% and 38.29 ± 1.66%, respectively. During all the fermentation process, it was possible to observe that the SFAs content was not affected by the fermentation process, and the concentration detected at the beginning of the fermentation was stable until the end. Conversely, MUFAs and PUFAs showed an opposite trend during all the fermentation process. In fact, as confirmed by statistical test with significances higher than 95%, MUFAs increased significantly, up to 47.28 ± 2.85%, while PUFAs decreased down to 26.34 ± 1.14%. The polyunsaturated decreasing represents an advantage for the final fermented product, since its shelf life could be extended [64,65].

**Table 2.** Fatty acids contents at different fermentation times.

|  | SFAs (%) | MUFAs (%) | PUFAs (%) |
|---|---|---|---|
| 0 h | 24.91 ± 1.51 (A) | 35.48 ± 2.04 (A) | 38.29 ± 1.66 (A) |
| 24 h | 23.99 ± 1.49 (A) | 36.44 ± 2.08 (A) | 36.18 ± 1.54 (A) |
| 48 h | 24.01 ± 1.45 (A) | 46.43 ± 2.21 (A) | 26.55 ± 1.47 (A) |
| 72 h | 23.34 ± 1.44 (A) | 47.99 ± 2.35 (A) | 26.56 ± 1.38 (B) |
| 96 h | 24.55 ± 1.41 (A) | 47.32 ± 2.49 (B) | 26.2 ± 1.29 (B) |
| 120 h | 24.68 ± 1.39 (A) | 47.02 ± 2.52 (B) | 26.39 ± 1.29 (B) |
| 144 h | 24.46 ± 1.29 (A) | 47.28 ± 2.85 (B) | 26.34 ± 1.14 (B) |

The values in a column with different letters are significantly different ($p \leq 0.05$).

As previously reported in literature [66,67], this fatty acid behavior can be ascribable to the yeast metabolism that degrades fat for single cell protein production [67] and for obtaining the energy for metabolic activities during fermentation process [68]. The fatty acids resulting from lipids degradation can be subsequently degraded through β-oxidation system in the yeast cells [67], resulting in the polyunsaturated reduction.

### 3.3. Cell Wall Insoluble and Soluble Sugars

Table 3 shows the percentage of the insoluble monosaccharides detected during all the fermentation process, while the soluble fraction composition is reported in Figure 1.

**Table 3.** Monosaccharide insoluble sugars compositions of waste cell walls during the fermentation process.

| Hours | Residue | Totals | Rhamnose | Fucose | Arabinose | Xylose | Mannose | Galactose | Glucose | Uronic Acid |
|---|---|---|---|---|---|---|---|---|---|---|
| 0 | 3.9 | 736.67 ± 36.8 | 1.34 ± 0.34 | 0.80 ± 0.12 | 75.69 ± 3.87 | 198.51 ± 20.85 | 29.67 ± 3.38 | 55.82 ± 5.61 | 268.25 ± 20.58 | 106.59 ± 9.65 |
| 24 | 2.8 | 645.01 ± 32.6 | 1.26 ± 0.26 | 0.84 ± 0.07 | 64.66 ± 3.68 | 174.95 ± 14.85 | 28.68 ± 1.86 | 44.09 ± 4.58 | 209.34 ± 16.55 | 97.62 ± 5.24 |
| 48 | 2.1 | 519.06 ± 25.8 | 1.34 ± 0.19 | 0.76 ± 0.06 | 59.78 ± 2.25 | 132.04 ± 33.62 | 15.24 ± 1.08 | 38.38 ± 1.61 | 148.55 ± 15.51 | 80.07 ± 4.25 |
| 72 | 1.9 | 387.53 ± 22.6 | 1.21 ± 0.23 | 0.63 ± 0.08 | 43.34 ± 3.17 | 97.28 ± 21.03 | 13.46 ± 1.64 | 22.65 ± 1.91 | 116.94 ± 21.62 | 57.25 ± 5.26 |
| 96 | 1.4 | 289.55 ± 26.6 | 1.01 ± 0.25 | 0.51 ± 0.04 | 35.03 ± 4.01 | 97.26 ± 6.3 | 13.24 ± 1.18 | 20.59 ± 2.48 | 85.23 ± 6.52 | 36.65 ± 3.14 |
| 120 | 1.2 | 273.73 ± 24.0 | 1.16 ± 0.20 | 0.50 ± 0.04 | 29.56 ± 1.68 | 97.06 ± 5.14 | 12.71 ± 0.95 | 17.08 ± 1.26 | 83.24 ± 6.24 | 32.21 ± 2.54 |
| 144 | 1.2 | 266.99 ± 27.3 | 1.15 ± 0.20 | 0.51 ± 0.04 | 29.71 ± 1.86 | 92.02 ± 6.01 | 12.52 ± 1.02 | 16.25 ± 1.58 | 80.78 ± 4.75 | 29.01 ± 3.69 |

Results are expressed as μg/mg anhydrous sugars in original sample. Residue, expressed in percentage, is the proportion of biomass recovered as alcohol insoluble residue (AIR).

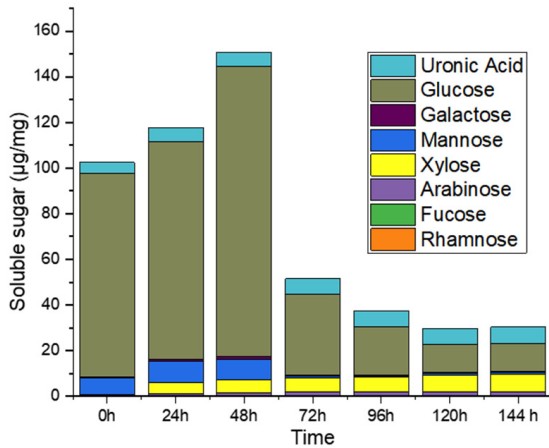

**Figure 1.** Monosaccharide soluble sugars compositions during the fermentation process. Results are expressed as μg/mg anhydrous sugars in original sample.

The main insoluble sugars resulting from the cell wall hydrolysis of AIR waste residues detected at the beginning of the fermentation were 268.25 ± 20.58 μg/mg (6.97% on dry matter) glucose (Glc), 198.51 ± 20.85 μg/mg (5.16% on dry matter) xylose (Xyl) and 106.59 ± 9.65 μg/mg (2.77% on dry matter) uronic Acid (UA), followed by 75.69 ± 3.87 μg/mg (1.97% on dry matter) arabinose (Ara), 55.82 ± 5.61 μg/mg (1.45%on dry matter) galactose (Gal) and 29.67 ± 3.38 μg/mg (0.77% on dry matter) mannose (Man), with smaller amounts of rhamnose (Rha) and fucose (Fuc). Whereas the main soluble sugars detected at the beginning were represented by Glc and Man, reaching concentrations of 89.25 ± 2.60 μg/mg (17.85% on dry matter) and 7.31 ± 0.89 μg/mg (1.46% on dry matter), respectively.

Observing the total insoluble sugars concentration during all the fermentation period, it can be noticed that the substrate was hydrolyzed since the early phases of the process, as a consequence of the enzymes' addition. In fact, a slight decrease in the insoluble fraction was recorded by 24 h. After 96 h, the total insoluble sugars decreased from 736.67 ± 36.8 μg/mg (19.15% on dry matter) down to 289.55 ± 26.6 μg/mg (2.70% on dry matter) and this concentration remained stable until the end of the process. This decreasing was mainly due to a decrease in Glc, Xyl, UA and Ara concentrations during the process, while the Rha and Fuc concentrations were stable throughout.

The total insoluble cell wall sugars decreasing was due to the enzyme addition, which enhanced the digestible carbohydrate while decreasing the content of nondigestible fiber, making the resulting substrate suitable as animal feed [69].

In the samples of digested materials, the insoluble sugars decrease was followed by an increasing concentration of soluble Glc, Man, Xyl, Ara and UA by the first 48 h.

In Figure 2 is reported the trend of soluble and insoluble sugars and the protein increasing during the fermentation process, calculated on dry matter bases, and expressed in percentages.

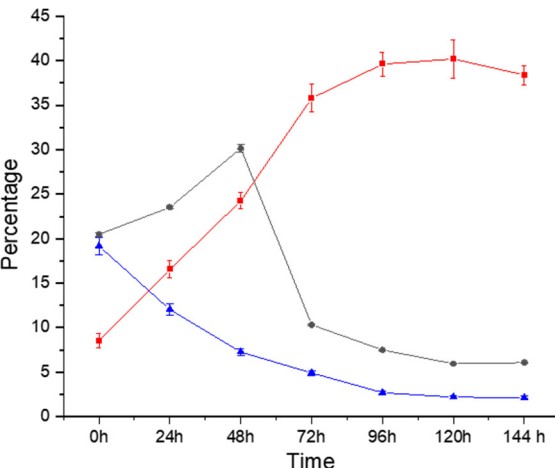

**Figure 2.** Percentage of protein, soluble and insoluble sugars detected during the fermentation process. Red square: crude protein; blue triangle: insoluble sugar; grey circle: soluble sugar. Results are expressed as percentage calculated on dry matter.

The highest concentration of soluble sugars was reached after 48 h (Figures 1 and 2), when the Glc concentration detected was 126.85 ± 2.75 µg/mg (21.14% on dry matter), followed by 5.62 ± 0.85 µg/mg (3.00% on dry mater) Xyl, 6.25 ± 0.88 µg/mg (2.50% on dry matter) UA, 8.82 ± 0.88 µg/mg (2.02% on dry matter) Man, 1.33 ± 0.01 µg/mg (0.84% on dry matter) Ara and 1.36 ± 0.45 µg/mg (0.60% on dry matter) Gal. The soluble sugar increasing was the result of the insoluble sugar percentage decreasing. In fact, at that stage, the total insoluble sugar decreased from 736.67 ± 36.8 µg/mg (19.15% on dry matter) down to 519.06 ± 25.8 µg/mg (7.27% on dry matter). Focusing the attention on the main fermentable sugar by yeast, glucose, it is possible to notice that its increasing in the medium (+3.29%) was lower in comparison with the decreasing observable in the insoluble fraction (−4.89%). This can be ascribable to its utilization by the yeast for growing and its replacement in the media due to the enzymatic hydrolysis [70].

As observed for the insoluble sugars fraction, the composition of soluble sugars also reached up a steady state after 72 h (Figure 1). In this fraction, in fact, the main fermentable sugars of the microorganism employed were almost totally used for growing, while the pentose sugars, mainly represented by xylose and arabinose, were left unused in the medium, since the yeast was unable to use the pentoses [21,38,42]. It is important to notice that, while mannose and galactose were not used during the first 48h of the process, after 72 h of fermentation their concentration in the medium was decreasing. This could be ascribable to the ability of *S. cerevisiae* of metabolize these sugars at low glucose concentration in the substrate [71]. At the end of the fermentation process, soluble sugars detectable in the media were represented by pentose and uronic acid, which increased in the substrate due to the enzymatic pectin hydrolysis and the corresponding increase in UA in the medium.

## 4. Conclusions

This study demonstrated an effective approach to use a multicomplex substrate, made up by different food waste, to set up a lab scale fermentation process allowing the performance of a simultaneous biovalorization of different food waste, without the necessity to separate the waste coming out from food process for getting an added value product. In particular, this study pointed out the possibility to use both animal and vegetable food waste together, by a simultaneous saccharification and fermentation process for obtaining single cell protein by *S. cerevisaie*, suitable as animal feed.

The final fermented product resulted was enriched in SCP, as a result of cell wall sugars' utilization for yeast growth, and it showed an adequate composition for animal

feed by encouraging the conversion of fish waste together with agricultural food waste in feed by using a low cost process.

Utilization of residues not only eliminates the disposal problems but also solves pollution associated problems.

Further studies are required to set up a scale up of the fermentation process for its proposal on an industrial scale, besides an economic feasibility evaluation also focused on the conversion of the resulting fermentation product into pellet and the evaluation of its effect on animal palatability and growth when used as feed.

**Author Contributions:** Conceptualization, A.T. and G.D.B.; methodology, A.F., A.A., A.G.P., G.D.B., V.L.T. and A.T.; formal analysis, A.F., A.A., A.G.P., G.D.B., V.L.T. and A.T.; investigation, A.F., A.A., A.G.P., G.D.B., V.L.T. and A.T.; data curation, A.F., A.A., A.G.P., G.D.B., V.L.T. and A.T.; writing—original draft preparation, A.F., A.A., A.G.P., G.D.B., V.L.T. and A.T.; writing—review and editing, A.T.; supervision, A.T.; All authors have read and agreed to the published version of the manuscript.

**Funding:** This research received no external funding.

**Institutional Review Board Statement:** Not applicable.

**Informed Consent Statement:** Not applicable.

**Data Availability Statement:** Not applicable.

**Conflicts of Interest:** The authors declare no conflict of interest.

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
