# Peer review of "Single Cell Protein Production through Multi Food-Waste Substrate Fermentation"

_fermentation, doi:10.3390/fermentation8030091_

Round 1

Reviewer 1 Report

The authors have submitted a manuscript in which they develop a process based on fermentation by Saccharomyces cerevisiae ATCC 36858 using fish waste and agricultural waste as substrate for obtaining single cell protein suitable as feed.

The topic of this paper is suitable for the scope of the Journal. The novelty could be explained better. Is this the first time that Saccharomyces cerevisiae was used for the fermentation of fish and agricultural waste? Are there similar papers in the literature?

The abstract could be improved. E.g., in Line 15 you write “this waste”, but it is not clear what waste you are referring. The novelty should be included in the abstract.

The Introduction is quite well presented. I just suggest adding some details about food waste (tons produced, actual use …)

Paragraph 2.1: Which are the proportions of fish and agricultural waste? What do you mean with “in a proportion of 20% (w/w) for each kind”?

Paragraph 2.3: Why did you use these conditions for the fermentation? Are these the optimal conditions for enzyme used? Please, specify.

Paragraph 2.5: What do you mean with “true protein content”? Maybe “total”?

In the tables use always “.” instead of “,”

Paragraph 3.1

Can you compare your results with other obtained in similar papers?

Paragraph 3.3

Could it be interesting to analyze the relative percentage of the total of each monosaccharide over time?

Again, can you compare your results with other obtained in similar papers?

Conclusions can be slightly improved. You have put here some interesting considerations on the novelty of the work, which instead could be better explained in the abstract and in the introduction. What tests would be needed to verify the use of these proteins as feed? What are the current limits for proposing this process on an industrial scale?

Other minor remarks:

Put always °C after a space

Uniform the spacing using ±

Revise English grammar, as there are some minor mistakes

E.g., line195 shows instead of shown

Reviewer 2 Report

In this paper, fish waste and agricultural waste were used for single cell protein production by Saccharomyces cerevisiae ATCC 36858, the single cell protein can reach up 40.19%, the soluble and insoluble sugars dropped down from 20.5% to 6.10% and 19.15% to 2.14% respectively, at the end of the fermentation process.

This paper, design appropriate, Materials and methods are adequately described, results and conclusions were clearly and detailed.

However, this paper slightly lacks of innovation.

I suggested accept after minor revision.

1、In paper title, “SCP” should be revised to Single cell protein.
